# Generation of an Open-Access Patient-Derived iPSC Biobank for Amyotrophic Lateral Sclerosis Disease Modelling

**DOI:** 10.3390/genes14051108

**Published:** 2023-05-18

**Authors:** Erin C. Hedges, Graham Cocks, Christopher E. Shaw, Agnes L. Nishimura

**Affiliations:** 1United Kingdom Dementia Research Institute, Maurice Wohl Clinical Neuroscience Institute, Institute of Psychiatry, Psychology and Neuroscience, King’s College London, 5 Cutcombe Rd., London SE5 9RT, UK; erin.hedges@kcl.ac.uk; 2Genome Editing and Embryology Core, King’s College London, London SE1 1UL, UK; graham.d.cocks@kcl.ac.uk; 3Centre for Brain Research, University of Auckland, 85 Park Road, Grafton, Auckland 1023, New Zealand; 4Blizard Institute, Neuroscience, Surgery and Trauma, Queen Mary University of London, 4 Newark Street, London E1 2AT, UK

**Keywords:** induced pluripotent stem cell, amyotrophic lateral sclerosis, disease modelling, biobank

## Abstract

Amyotrophic lateral sclerosis (ALS) is a progressive neurodegenerative disease affecting the upper and lower motor neurons, causing patients to lose control over voluntary movement, and leading to gradual paralysis and death. There is no cure for ALS, and the development of viable therapeutics has proved challenging, demonstrated by a lack of positive results from clinical trials. One strategy to address this is to improve the tool kit available for pre-clinical research. Here, we describe the creation of an open-access ALS iPSC biobank generated from patients carrying mutations in the *TARDBP*, *FUS*, *ANXA11*, *ARPP21*, and *C9ORF72* genes, alongside healthy controls. To demonstrate the utilisation of these lines for ALS disease modelling, a subset of *FUS*-ALS iPSCs were differentiated into functionally active motor neurons. Further characterisation revealed an increase in cytoplasmic FUS protein and reduced neurite outgrowth in *FUS*-ALS motor neurons compared to the control. This proof-of-principle study demonstrates that these novel patient-derived iPSC lines can recapitulate specific and early disease-related ALS phenotypes. This biobank provides a disease-relevant platform for discovery of ALS-associated cellular phenotypes to aid the development of novel treatment strategies.

## 1. Introduction

Since their discovery in 2007, human induced pluripotent stem cells (iPSCs) have broadened our understanding of basic biology and transformed the field of developmental biology [1]. iPSCs harbour similar properties to embryonic stem cells and can be guided to differentiate into various specialised cell types with high levels of purity, but they are derived from somatic cells. iPSC-derived neurons can be generated in as little as three days using doxycycline-inducible transcription factor systems or in two weeks using chemically defined conditions [2,3]. In addition, advances in 3D organoid and co-culture systems mean we can model diseases with greater complexity than ever before in human cells [4].

iPSCs hold great promise for drug screening and personalised medicine [5] and can be derived from patients with neurodegenerative diseases such as Alzheimer’s disease, Parkinson’s disease, and amyotrophic lateral sclerosis (ALS) [6,7,8]. iPSCs are readily accessible, have unmatched capabilities when modelling specific cell types in vitro, and are not burdened with the ethical concerns typically associated with human embryonic stem cells [5]. In addition, the generation of genetically modified iPSC lines increases the versatility of stem cell models of complex genetic disorders. This can be achieved with CRISPR/Cas9 technology by correcting genetic mutations in patient-derived lines or inserting mutations into well-characterised healthy control lines [9]. Patient-derived iPSC lines can give rise to a large spread in experimental data due to the inherent genetic variability present across cell lines derived from different individuals. This is somewhat avoided with the generation of isogenic cell lines with CRISPR/Cas9 technology. However, careful considerations must be made when utilising these models, such as analysis of off- and on-target effects [10,11]. These can cause variability in CRISPR/Cas9-generated iPSC lines, resulting in cellular phenotypes that are not necessarily associated with the mutation being modelled. Further, it is important to instil a measured interpretation of phenotypes that have been observed within a single genetic background when extrapolating experimental results from isogenic lines. Combined analysis of CRISPR/Cas9-generated and patient-derived iPSC lines may be essential for the elucidation of real cellular phenotypes from non-specific signals that can arise from clonal and experimental variability. Therefore, the generation of iPSC lines from patients with a range of genetic disease risk factors will broaden the scope of pre-clinical research and improve therapeutic targeting for a larger patient cohort.

ALS is a neurodegenerative condition which affects the upper and lower motor neurons [12], with a global mortality rate of approximately 30,000 patients each year [13]. In the UK alone, the total number of newly diagnosed ALS cases is estimated to rise from 1701 in 2020 to 2635 in 2116 [14]. ALS has no cure, and over 50 clinical trials (CTs) have failed over the past 25 years [13,15]. Currently, Riluzole is the only disease-modifying drug for ALS, extending survival by 6–19 months, and it received commercial authorisation in Europe in 1996 [16]. A retrospective study found that Riluzole-mediated survival occurs in the last clinical stage of ALS, indicating that increased lifespan occurs during the disease stage where symptoms are most severe [17]. Co-treatment with Riluzole and the tyrosine kinase inhibitor Masitinib slowed the rate of functional decline compared to Riluzole plus placebo in a phase 2/3 clinical trial [18], indicating that combination therapy may be beneficial in ALS. The drug Edaravone received approval from the FDA in the USA in 2017 [19] but did not yield a positive outcome in a multicentre clinical trial in Italy [20]. The lack of viable and novel therapeutics for ALS indicates that more research and better drug–target identification are essential.

ALS is characteristically complex and heterogeneous which has likely contributed to the overriding failure of ALS-CTs. ALS subclassification and stratification may increase the success rate of CTs [21,22], which might be achieved through patient phenotyping and genotyping. Screening patients for known genetic causes of ALS is particularly relevant when developing treatment strategies that target specific genomic aberrations, such as gene therapies, including antisense oligonucleotides (ASOs) [23]. The first ASO for treatment of *SOD1*-ALS, Tofersen, was recently approved by the FDA; a milestone in drug development for ALS [24]. Additional data suggest that ASOs targeting specific genetic mutations in *SOD1*, *C9ORF72*, *FUS*, and *ATXN2* hold potentially beneficial clinical outcomes [25], and patient-derived cells can be used to test novel therapeutics such as ASOs. Neurons and glia remarkably recapitulate key pathological features associated with ALS including increased neuronal death and mislocalisation of disease-associated proteins to the cytoplasm [26,27,28,29]. These cellular phenotypes represent useful markers of drug efficacy in disease-relevant cell types, indicating how improving the availability and variety of patient-derived iPSCs may enhance the translational impact of research.

Patient-derived cell biobanks allow us to determine disease-related phenotypes and test the toxicity and effectiveness of novel compounds. Such a biobank was created in 2003 by The Motor Neurone Disease Association, named The UK MND Collections. This biobank contains more than 3000 blood samples including lymphoblastoid cell lines (LCLs) and peripheral blood lymphocytes (PBLs) from healthy volunteers and ALS patients, alongside clinical and epidemiological information [30]. LCLs are immortalised cell lines derived from peripheral B lymphocytes infected with Epstein–Barr virus (EBV) [31]. LCLs are an excellent source of DNA and are a useful tool for in vitro experimentation; their stable genome and transcriptome properties, inexpensive maintenance, and easy manipulation make LCLs great value for disease research [32]. However, LCLs do not represent the cell types affected by disease when modelling neurodegenerative conditions such as ALS. To address this, we repurposed The UK MND Collections as a resource for generating patient-derived iPSC lines.

Here, we describe the generation of this ALS-iPSC biobank alongside proof-of-concept data generated with *FUS*-ALS patient-derived iPSC-neurons to demonstrate the prevalence of disease-specific early phenotypes in patient-derived cells. Thirty-five iPSC lines were generated as part of The UK MND Collections, which can be openly accessed through The Motor Neurone Disease Association.

## 2. Materials and Methods

### 2.1. LCL and PBL Culture

LCLs were cultured in upright non-adherent T25 tissue culture flasks (Nunc) at 37 °C and 5% CO_2_. LCL media consisted of RPMI 1640 (Thermo Fisher, Waltham, MA, USA), supplemented with 1× GlutaMAX™ (Thermo Fisher), and 20% fetal bovine serum (Thermo Fisher). Erythroblasts were enriched in mixed PBL cultures following a previously published protocol [33]. Briefly, PBLs were cultured in erythroblast media containing StemSpan™ SFEM (Stem Cell Technologies, Vancouver, BC, Canada), 50 ng/mL recombinant human stem cell factor (R&D Systems, Minneapolis, MN, USA), 1 μM Dexamethasone (Sigma-Aldrich, St. Louis, MO, USA), 40 ng/mL IGF1 (Miltenyi Biotec, Bergisch Gladbach, Germany), 10 ng/mL Interleukin 3 (Miltenyi Biotech), 2 U/mL human erythropoietin (R&D Systems), and 10 μg/mL Gentamicin (Thermo Fisher). Magnetic beads labelled with a CD71 antibody were then used to enrich for erythroblasts, following the manufacturer’s instructions (Miltenyi Biotec).

### 2.2. Generation and Maintenance of iPSCs

iPSCs were reprogrammed from LCLs and PBLs as previously reported [34,35,36]. iPSCs were generated by nucleofection of 1 × 10^6^ cells with plasmids expressing *OCT3/4*, *L-MYC*, *KLF4*, *SV40LT*, *LIN28*, *SOX2,* and *shRNA-p53* (addgene codes: #20927, #27077, #27080, #27078) with the Amaxa™ Human B Cell Nucleofector™ Kit (Lonza, Basel, Switzerland). After electroporation, cells were transferred onto a feeder layer of inactivated mouse embryonic fibroblasts in LCL media. Cells were maintained in LCL media for five days, then media was transitioned to reprograming media (RM) (DMEM/F12 with GlutaMAX™ (Thermo Fisher), 1× non-essential amino acids (Thermo Fisher), 1× N2 Supplement (Thermo Fisher), 1× B27™ Supplement (Thermo Fisher), 0.1 µM 𝛽-mercaptoethanol (Thermo Fisher), 100 ng/mL bFGF (PeproTech, Cranbury, NJ, USA), 1000 U/mL hLIF (Merck Millipore, Burlington, MA, USA), 0.5 µM PD-0325901 (BioVision, Milpitas, CA, USA), 3 µM CHIR99021 (Cayman Chemical, Ann Arbor, MI, USA), 10 µM HA-100 (Santa-Cruz, Santa Cruz, CA, USA), and 0.5 µM A83-01 (BioVision)). RM was then replaced every 1–2 days. iPSC colonies were identified around two weeks later, and media was transitioned to Essential 8™ Flex medium (Thermo Fisher) until colonies were large enough for manual picking. Established colonies were cultured on Geltrex™ (Thermo Fisher) in Essential 8™ Flex. Two LCL-derived iPSC lines were derived using CytoTune™ iPS 2.0 Sendai Reprogramming Kit (Thermo Fisher), following the manufacturer’s protocol, with an additional spinfection step of 2250 RPM for 90 min in a large benchtop centrifuge immediately after the virus was added to LCLs. LCLs were maintained in LCL media for 48 h, with a media change at 24 h. LCLs were transferred onto inactivated mouse embryonic fibroblasts, maintained in LCL media until day 5, and then transitioned to RM as described above. iPSCs were routinely screened to ensure the absence of contamination with mycoplasma before cryopreservation using the MycoAlert^®^ Mycoplasma Detection Kit (Lonza, Basel, Switzerland).

### 2.3. iPSC Characterisation

iPSCs were subject to quality control (QC) experiments, including pluripotency immunocytochemistry and embryoid body assay (see below). Patient-derived lines were sequenced at the mutation region by sending PCR products and primers to Source BioScience (London, UK). Genomic integrity was assessed by G-band or digital karyotyping: G-band karyotyping was outsourced to TDL Genetics (London, UK) or to the Genome Editing and Embryology Core (King’s College London, UK), and digital karyotyping was completed with KaryoStat™ Karyotyping Service (Thermo Fisher). To confirm the loss of EBV genes, iPSC lines were serially passaged, and genomic DNA was screened for EBV genes with PCR. DNA was extracted using the DNeasy Blood and Tissue Kit (Qiagen, Hilden, Germany), and PCR was performed using primers targeting EBV genes (EBNA2, LMP1, BZLF and oriP) [34] and the housekeeping gene SDHA, using Q5^®^ High-Fidelity 2× Master Mix (NEB, MA, USA) with 35 cycles of 95 °C for 30 s, 61 °C for 30 s, and 72 °C for 30 s. PCR products were separated by gel electrophoresis in 4% agarose gels with 1% ethidium bromide, then visualised and photographed inside a UV transilluminator. STR profiling was outsourced to Source BioScience (London, UK); 16 STR loci were analysed and matched between iPSCs and parent LCL or PBL lines.

### 2.4. Embryoid Body Assay

iPSCs were dissociated with Versene and transferred to poly-HEMA (Sigma) coated plates in Essential 8™ Flex medium (Thermo Fisher) with 10 μM Rock inhibitor (BioVision). The next day, media was changed to embryoid body media ((KnockOut™ DMEM (Thermo Fisher), 10% knock-out serum replacement (Thermo Fisher), 5% fetal bovine serum (Thermo Fisher), 1× non-essential amino acids (Thermo Fisher), 12 ng/mL hLIF (Merck Millipore), and 55 µM 𝛽-mercaptoethanol (Thermo Fisher)) after which media was replaced every 2–3 days. After one week, embryoid bodies were transferred to glass coverslips coated with 0.1% gelatine for spontaneous differentiation. Media was changed every 2–3 days for three weeks until fixation.

### 2.5. Motor Neuron Differentiation

Motor neurons were generated using a pre-established protocol [37]. First, neuroepithelial cells were generated by culturing iPSCs in N2B27 media (50% Neurobasal™, 50% DMEM:F12, 0.5× N2 supplement, 0.5× B27™ supplement, and 1× GlutaMAX™ (all Thermo Fisher)) supplemented with 3 μM CHIR99021 (Tocris, Bristol, UK), 2 μM dorsomorphin (Tocris), and 2 μM SB431542 (Tocris) for four days. Neuroepithelial cells were expanded and differentiated into motor neuron progenitors with N2B27 media supplemented with 0.1 μM retinoic acid (RA) (Sigma-Aldrich) and 0.5 μM Purmorphamine (Tocris) for another two days. On day six, CHIR99021, dorsomorphin and SB431542 were withdrawn, and cells were cultured with RA and dorsomorphin for an additional six days. Media was transitioned to maturation media (BrainPhys™ Neuronal Culture Media (Stem Cells technologies, Vancouver, BC, Canada), 0.5× N2 Supplement (Thermo Fisher), 0.5× B27™ Supplement (Thermo Fisher), 10 ng/mL BDNF (PerpoTech), and 10 ng/mL GDNF (Peprotech)) including 0.1 μM Compound E (Calbiochem, San Diego, CA, USA) for the first three days to induce terminal differentiation via Notch inhibition.

### 2.6. Immunocytochemistry

Cells were fixed with 4% paraformaldehyde for 15 min at room temperature, washed with PBS, permeabilised in 0.1% TritonX-100 for 15 min, and blocked with 10% normal donkey serum (Sigma-Aldrich) for one hour. Primary antibodies were diluted in 5% donkey serum and incubated overnight at 4 °C (OCT3/4 (Santa Cruz Biotechnologies, Dallas, TX, USA), Nanog (Abcam, Cambridge, UK), SMA (Abcam), AFP (R&D Systems, Minneapolis, MI, USA), OLIG2 (Millipore), NKX6.2 (Millipore), TUJI (Sigma-Aldrich), Islet 1 (BD bioscience), Hb9 (Developmental studies hybridoma bank), and FUS (Novus, St. Louis, MI, USA)). The next day cells were washed three times with PBS and incubated with anti-donkey Dylight™ secondary antibodies (Thermo Fisher) diluted 1:400 in 5% donkey serum for one hour at room temperature. Nuclei were stained with 1.25 μg/mL Hoechst (Thermo Fisher) for 5 min, washed with PBS three times, and mounted with FluorSave™ Reagent (Merck Millipore).

### 2.7. Imaging and Analysis

Images were acquired with a Leica CYR5000 light microscope (Leica Microsystems, Wetzlar, Germany), a Leica TCS-SP5 microscope (Leica Microsystems, Germany), or an Opera Phenix^®^ High-Content Screening System (PerkinElmer, Waltham, MA, USA). Motor neuron quantification was performed in the Columbus™ Image Data Storage and Analysis system (PerkinElmer, USA). Approximately 8–10 fields from two separate wells of a 96-well plate were quantified in three biological replicates. Neurite outgrowth was quantified in individual neurons using the ImageJ plug-in NeuriteTracer.

### 2.8. Calcium Imaging

Motor neurons were cultured in 96-well plates and aged for 108 days. Cells were incubated with 2 mM Fluo4-AM in an external solution (145 mM NaCl, 2 mM KCl, 5 mM NaHCO_3_, 1 mM MgCl_2_, 2.5 mM CaCl_2_, 10 mM glucose, and 10 mM Na-HEPES (pH 7.25)) and 0.02% Pluronic-F27 (Thermo Fisher) for 15 min at 37 °C. Subsequently, neurons were rinsed in fresh external solution for another 15 min at 37 °C. Live image collection was performed with an Opera Phenix^®^ High-Content Screening System with a 20× water objective. Data were collected for 2 min, with one image taken every 2.8 s, and data were processed in ImageJ. Spontaneous calcium fluctuations were calculated as relative Fluo4-AM fluorescence intensity normalised to the background (F-F0/F0) across 10 regions of interest (ROI).

## 3. Results

### 3.1. Generation of iPSCs from LCLs and PBLs

Twenty-four novel iPSC lines were generated from a selection of healthy control and ALS patient LCLs, and five iPSC lines were derived from PBLs (Table 1). An additional six patient-derived LCL-iPSC lines also constitute part of the UK MND Collections and have been reported [36]. Across the collection and within the ALS group, LCL-derived iPSCs were generated from five patients with mutations in *ANXA11* (2× G38R, 2× D40G, 1× R235Q), three with *TARDBP* mutations (1× M337V, 1× G348V, 1× N378D), four with *C9ORF72* GGGGCC intronic expansions, three with mutations in *FUS* (1× R519E, 1× R521H, 1× R522G), and four with mutations of unknown significance in *ARPP21* (3× P529L, 1× P713L). The remaining line is derived from a sporadic ALS patient with no known genetic mutation. An additional ten control lines were generated from healthy individuals (5× male, 5× female). Five PBL-derived iPSC lines were generated from the same donor cohort, including two healthy control and three ALS patient-derived lines: Two with mutations in *TARDBP* (1× G348V, 1× N378D) and one with a mutation of unknown significance in *ARPP21* (1× P529L). Healthy donor and ALS patient demographic information for the newly generated lines is included in Table 1.

### 3.2. Characterisation of Newly Derived iPSCs

Newly generated LCL- and PBL-derived iPSC lines were subject to standard QC testing. Example data are included in Figure 1, and individual QC data are available alongside cell lines. A summary of QC results for all cell lines is included in Table 2. All iPSCs showed typical morphology with small round cells with large nuclei growing in defined colonies (Figure 1B). Immunocytochemistry targeting the pluripotency markers OCT3/4 and Nanog indicated the expression of stem cell-specific proteins in iPSCs (Figure 1A). The embryoid body assay was included as an additional measure of pluripotency: iPSCs were allowed to spontaneously differentiate, and cultures were probed for cells originating from the three layers of the blastocyst. Embryoid bodies were immunolabelled for the mesodermal protein smooth muscle actin (SMA), the endodermal marker alpha-fetoprotein (AFP), and 𝛽3-Tubulin was used to identify cells derived from the ectoderm (Figure 1C).

Cell line identity was confirmed using STR profiling to ensure that daughter iPSC lines correctly matched parent LCL or PBL genetic profiles (in line with patient data protection). In addition, ALS patient-derived lines with known genetic mutations were directly sequenced and their genotypes were verified (Figure 1F). To confirm the loss of EBV genes in LCL-derived iPSCs, clonal lines were serially passaged, and genomic DNA preparations were interrogated for the persistence of EBV genes (Figure 1G). Where possible, if EBV genes were still present in iPSC lines beyond passage 30, the clonal line was discarded, and a new iPSC clone from the same donor was selected. The serial passaging in this QC step had the potential to introduce cell line abnormalities, and so was completed prior to any pluripotency or genomic analyses.

Measures of genome stability included either G-band karyotyping (Figure 1D) or digital karyotyping (Figure 1E). G-band karyotyping was initially included to confirm the absence of gross chromosomal changes that might arise during reprogramming. This was extremely labour intensive and low –throughput; therefore, KaryoStat™ digital karyotyping was subsequently included to circumvent these issues. KaryoStat™ was deemed a suitable alternative as although it cannot detect balanced translocations, it offers an increased indel resolution compared to G-banding and is able to detect culture mosaicism with a limit of 30%. Additionally, if indels are detected in cell lines, the affected loci are identified, and so the potential consequences of structural changes can be assessed based on the functions of affected genes and any known disease associations.

### 3.3. Phenotypic Screening

To investigate whether iPSC lines generated for this biobank can recapitulate key pathological features of ALS, we performed a preliminary analysis of iPSCs derived from two *FUS*-ALS patients. iPSCs carrying *FUS* R521H and R522G and two control lines were differentiated into motor neurons using small molecule mediated differentiation. iPSCs were first differentiated into OLIG2-positive motor neuron progenitors via an intermediate neuroepithelial stage (Figure 2A,B). Terminal differentiation was achieved by Notch inhibition, giving rise to neuronal cultures with ~70% *β*3-Tubulin positive cells and ~50% Islet 1 positive motor neurons (Figure 2B–D).

Motor neurons were cultured for 108 days and assessed for spontaneous calcium fluctuations as an indirect measure of synaptic activity. All lines presented with calcium transients, indicating the functional activity of neurons (Figure 2E,F). *FUS* R521H and R522G and control motor neurons were immunolabelled for FUS protein on day 21 of differentiation (Figure 3A), indicating a relative increase in FUS protein in the cytoplasm in patient-derived lines compared to controls (Figure 3B). Neurite outgrowth was assessed in young motor neurons on day 21 of differentiation, revealing a decrease in total neurite length in *FUS* patient-derived lines compared to controls (Figure 4).

## 4. Discussion

Thirty-five new iPSC lines derived from patients with ALS and healthy controls have been generated and characterised. These iPSCs were derived from patients with mutations in the *FUS*, *C9ORF72*, *TARDBP*, *ARPP21*, and *ANXA11* genes, and from one sporadic patient. LCLs and PBLs from The UK MND Collections were utilised as a resource for the generation of this biobank.

LCLs are B lymphocytes that have been immortalised by infection with EBV, a lymphotropic herpesvirus [38]. In the majority of latent human infections, EBV exists episomally in the nucleus [39], however, integration into the host genome can occur in cases of Burkitt Lymphoma and other malignancies [40,41,42]. Other EBV-associated diseases, such as mononucleosis, are not typically associated with host genome integration, and in many cases EBV infection does not cause disease [39]. One of the EBV elements, EBNA-1, influences the chromatin architecture of infected cells, creating an “open” chromatin state [43], which may facilitate transcription factor activation and iPSC reprogramming in infected LCL lines [44]. One remarkable characteristic of iPSCs generated from LCLs is that the EBV elements are lost after passaging (Figure 1G) [34,35]. The mechanism by which the EBV elements are lost is yet to be completely understood; we hypothesise that these iPSCs are derived from individual lymphoblastoid cells where viral genes have not integrated, and EBV episome loss from explanted nasopharyngeal carcinoma cells has been reported [45]. However, if and how viral episomes are lost from iPSCs is undetermined.

Loss of EBV genes from iPSCs is a reproducible phenomenon, as indicated here where most iPSCs derived from ALS patients and control LCLs lost EBV genes before passage 30 (Figure 1G) [34,35]. However, EBV genes could be detected in genomic DNA extracts beyond passage 30 in approximately 25% of the screened iPSC clones. In these instances, it is possible that the EBV elements had integrated into the genome of the original lymphoblastoid, and thus the daughter iPSC. Subsequent analysis of EBV integration was not conducted in these instances. Prior infection with EBV was recently associated with multiple sclerosis, indicating that the presence of EBV genes might contribute to motor neuron pathophysiology [46]. To circumvent any possible confounding effects of EBV elements on cellular phenotypes, clonal iPSC lines expressing EBV genes beyond passage 30 were discarded, and a new clone was selected for characterisation. This was unexpected and time-consuming, which should be considered when utilising LCLs as a resource for iPSC reprogramming in future studies. Screening for EBV or other viral genes is not necessary when iPSCs are derived from primary cells such as PBLs. No qualitative differences were observed in either the success of iPSC characterisation or in routine iPSC culture in lines derived from each cell type. This suggests that where both materials are available when generating iPSCs from a desired genotype, PBL-derived iPSCs might be a more time- and cost-effective choice. No thorough comparison of LCL versus PBL iPSCs or differentiated cell types is reported here, and additional phenotyping will be necessary to solidify this assertion.

The selection of patient tissue for reprogramming was influenced by the lack of iPSC lines representing certain genotypes for ALS research. In particular, cell lines with mutations in *ANXA11* and *ARPP21* have not been previously reported, apart from some *ANXA11* patient-derived lines that constitute part of this same collection [36]. Mutations in *ANXA11* have a proven association with ALS [47,48,49], and the generation of these cell lines will be an important step in elucidating the role of the corresponding protein, Annexin A11, in motor neuron pathogenesis. Mutations in *ARPP21* have been identified in ALS patients [50], but the significance of these mutations is unconfirmed [51,52]. The utilisation of these newly generated lines will be essential in confirming the true contribution of *ARPP21* to the ALS genetic landscape. Concurrently, if mutations in *ARPP21* do not contribute to ALS risk, these cell lines represent sporadic and familial ALS patients with an unknown genetic burden.

Additional lines were generated from patients with well-established genetic associations, namely *TARDBP*, *FUS*, and *C9ORF72*. Multiple iPSC lines with mutations in these genes have been instrumental in progressing our understanding of the cellular pathologies associated with ALS, and these data have been excellently reviewed elsewhere [8,53,54,55]. iPSC lines generated for this biobank will add to this resource, providing greater opportunities for the identification of cellular pathology and pre-clinical validation of new therapeutics. Newly generated *FUS*-ALS iPSC-derived neurons mirror the cytoplasmic FUS mislocalisation seen in ALS post-mortem tissue and other *FUS*-ALS disease models, including other iPSC-derived models (Figure 3) [56,57,58,59]. Neurite outgrowth or branching defects have been observed in previously established *FUS*-iPSC models, showing variable results, with some reports of increased branching and length in *FUS* lines [60,61], and others indicating reduced complexity and outgrowth [62]. In addition, increasing the number of reliable control lines is essential when utilising patient-derived iPSCs with variable genetic backgrounds. Therefore, ten healthy control lines were generated from donors above the age of 60 years, ranging from 61–84 years of age and derived from males and females presenting with no neurological or health condition at the time of blood collection.

In summary, we have generated an open-access iPSC biobank, including multiple lines generated from ALS patients and controls. These can be accessed through The Motor Neurone Disease Association (https://www.mndassociation.org/research/for-researchers/resources-for-researchers/ukmndcollections/ (accessed on 10 May 2023)) alongside QC data for each cell line. In addition, uncharacterised clones from the same cell lines may be available to those interested in the comparison of clonal lines (Table 2). As an example of utilisation of these lines, we have shown that motor neurons derived from *FUS*-ALS patients recapitulate key pathological features observed in ALS patient tissue and other *FUS* models. Hence, these newly generated lines will aid ALS disease modelling, contributing to pre-clinical research on developing novel therapeutics.

## Figures and Tables

**Figure 1 genes-14-01108-f001:**
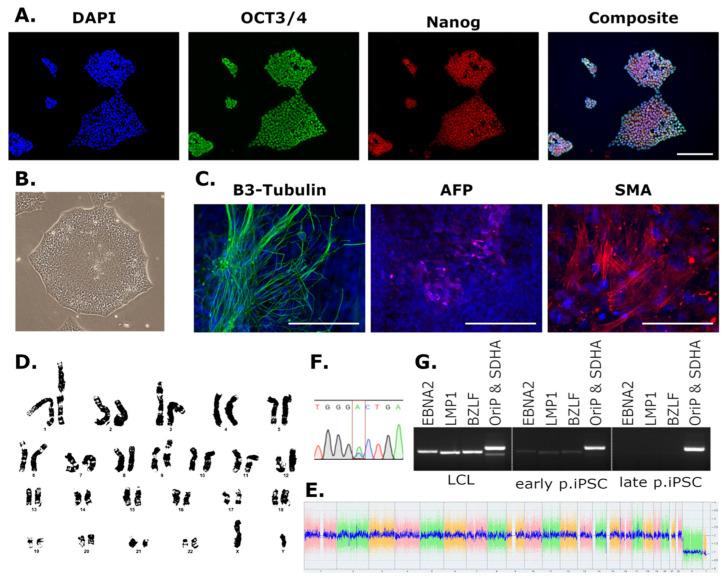
iPSC characterization (**A**) iPSCs positive for pluripotency markets OCT3/4 and Nanog measured by immunocytochemistry. (**B**) iPSCs show typical iPSC morphology. (**C**) Embryoid bodies immunolabelled for 𝛽3-Tubulin (ectoderm), alpha-fetoprotein (AFP, endoderm), and smooth muscle actin (SMA, mesoderm). (**D**) Karyograph from G-band karyotyping. (**E**) Example of KaryoStat+ results with no genetic abnormalities. (**F**) Sanger sequencing of mutation region in the patient line SP3154, harbouring a *TARDBP* G348 V mutation (GGC→GTC, reverse strand shown GCC→GAC). (**G**) PCR detecting EBV elements (*EBNA1*, *LMP1*, *BZLF*, and *OriP*) in LCL and early passage iPSC DNA, which are lost in late passage iPSC DNA (*SDHA* = housekeeping). (**A**–**D**,**G**) are QC data for the control line LC0209. (**E**,**F**) relate to the *TARDBP* line SP3154. Scale bars = 100 μm.

**Figure 2 genes-14-01108-f002:**
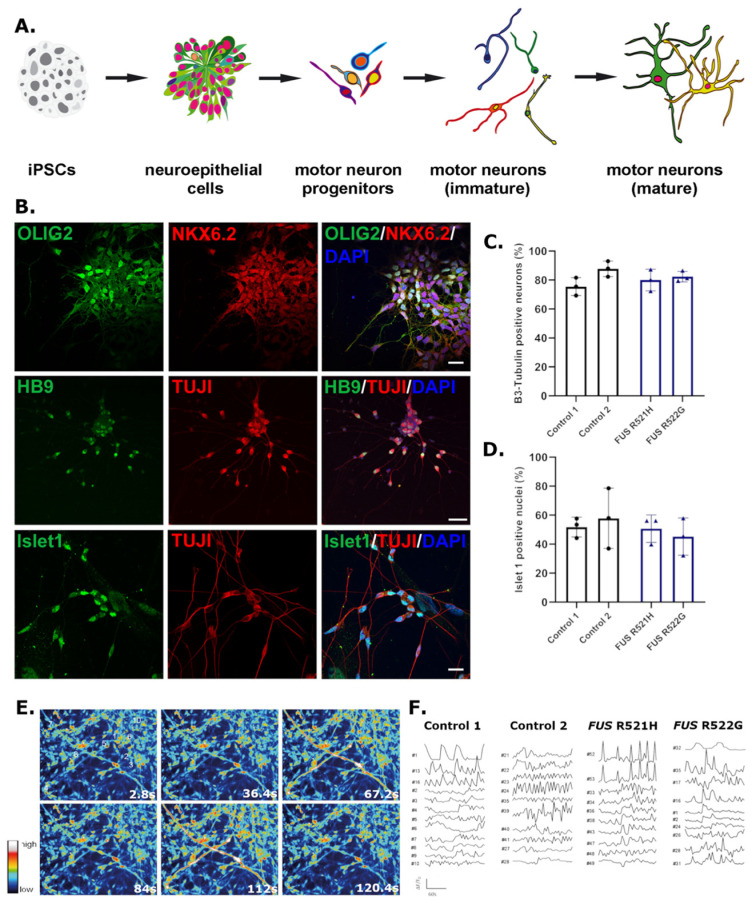
iPSCs differentiate into functional motor neurons (**A**) Schematic of motor neuron differentiation protocol including intermediate neuroepithelial and motor neuron progenitor stages. (**B**) Representative images of iPSC-derived motor neuron progenitors (top panel) and motor neurons (middle and bottom panels) immunolabelled for cell lineage-specific markers OLIG2, NKX6.2 (progenitors) and HB9, and Islet 1 (motor neurons), co-labelled with the neuronal marker 𝛽3-Tubulin (TUJ1). (**C**) Percentage of cells positive for the neuronal marker 𝛽3-Tubulin in two control and two *FUS*-ALS lines. (**D**) Percentage of neurons positive for the motor neuron marker Islet 1 in two control and two *FUS*-ALS lines. (**C**,**D**) Values are mean ± SD. (**E**) Still frames from spontaneous calcium live imaging in 108-day old motor neurons. (**F**) Spontaneous calcium activity quantified as ∆F/F0; each trace represents a different ROI in (**E**). Statistical analysis (**C**,**D**): One-way ANOVA with Tukey’s multiple comparisons test (*p* > 0.05), *n* = 3 (separate motor neuron inductions). Scale bars = 25 μM (OLIG; Islet 1) and 50 μM (HB9).

**Figure 3 genes-14-01108-f003:**
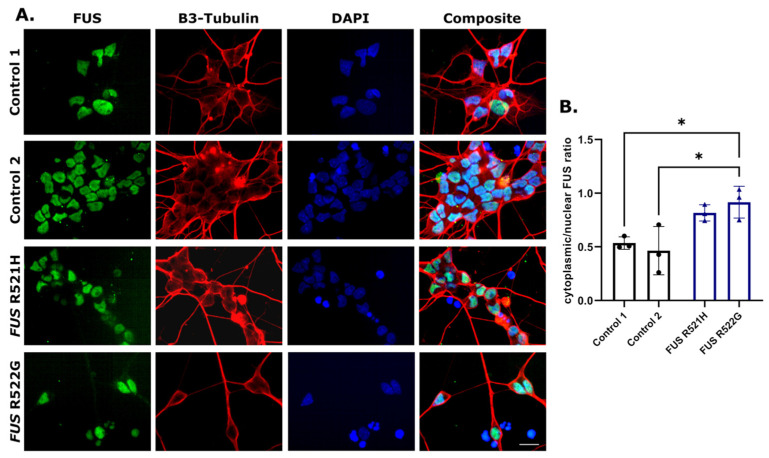
FUS protein mislocalises to the cytoplasm in FUS-ALS patient motor neurons (**A**) Two control and two FUS-ALS motor neuron cultures were immunolabelled for FUS and 𝛽3-Tubulin on day 21 of differentiation. (**B**) Cytoplasmic/nuclear FUS protein intensity ratio was increased in FUS-ALS motor neurons compared to control (* *p* < 0.05). Values are mean ± SD. Statistical analysis: One-way ANOVA with Tukey’s multiple comparisons tests, *n* = 3 (separate motor neuron inductions). The total number of analysed neurons was control 1 (8736), control 2 (7299), *FUS* R521H (6574), and *FUS* R522G (6486). Scale bar = 50 μm.

**Figure 4 genes-14-01108-f004:**
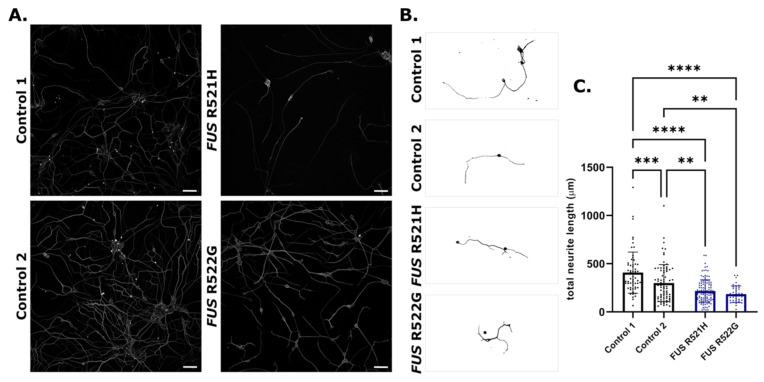
FUS-ALS motor neurons display reduced neurite outgrowth. (**A**) Representative images from an Opera Phenix^®^ High-Content Screening System of iPSC-derived motor neurons on day 21 of differentiation, immunolabelled for 𝛽3-Tubulin. (**B**) Example neurite traces of individual neurons from control and FUS-ALS motor neurons. (**C**) Total neurite length per neuron was reduced in FUS-ALS motor neurons compared to control (** *p* < 0.01; *** *p* < 0.001; **** *p* < 0.0001). Values are mean ± SD. Statistical analysis: One-way ANOVA with Tukey’s multiple comparisons tests, *n* = 3 (separate motor neuron inductions). The total number of analysed neurons was control 1 (130), control 2 (158), *FUS* R521H (238), and *FUS* R522G (82). Scale bars = 50 μm.

**Table 1 genes-14-01108-t001:** Donor information.

MNDA Cell Line ID	Status	Gene	Mutation	Age at Collection	Sex
LC0501	Control	−	−	69	Female
SNC0106	Control	−	−	71	Female
BLI0083	Control	−	−	72	Female
BC6325	Control	−	−	83	Female
SC3709	Control	−	−	84	Female
LNH0108	Control	−	−	61	Male
LCA0042	Control	−	−	61	Male
LC0209	Control	−	−	71	Male
BC6055	Control	−	−	74	Male
SC3602	Control	−	−	64	Male
LP0584	Familial	*ANXA11*	D40G	75	Female
LP0582	Familial	*ANXA11*	D40G	76	Female
SMA0020	Familial	*ANXA11*	G38R	51	Male
LP0663	Familial	*ANXA11*	G38R	63	Male
LP0039	Sporadic	*ANXA11*	R235Q	66	Female
BP6184	Sporadic	*ARPP21*	P713L	59	Male
SP3185	Sporadic	*ARPP21*	P529L	45	Male
LP0225	Familial	*ARPP21*	P529L	41	Female
SP3277	Familial	*ARPP21*	P529L	45	Male
SMA0078	Familial	*TARDBP*	N378D	60	Male
SP3068	Familial	*TARDBP*	M337V	59	Male
SP3154	Familial	*TARDBP*	G348V	58	Male
BOX0029	Familial	*C9ORF72*	G_4_C_2_ intronic expansion	58	Female
LPO0036	Familial	*C9ORF72*	G_4_C_2_ intronic expansion	64	Male
BP6021	Sporadic	*C9ORF72*	G_4_C_2_ intronic expansion	38	Female
BP6204	Sporadic	*C9ORF72*	G_4_C_2_ intronic expansion	45	Female
LP0393	Familial	unknown	-	71	Female
LP0048	Familial	*FUS*	R522G	29	Male
LP0051	Familial	*FUS*	Q519E	52	Male
LP0168	Familial	*FUS*	R521H	44	Female

**Table 2 genes-14-01108-t002:** iPSC characterisation information.

MNDA Cell Line ID	iPSC Derivation Method	Parent Cell Type	Number of Clones Frozen	Characterisation					
				Pluripotency ICC	Embryoid Body Assay	Karyotyping	EBV Loss (Passage of Negative Stocks)	Confirmation of ALS Mutation	STR Profilling
LC0501	P	LCL	8	✓	✓	G	P30	n/a	✓
SNC0106	P	LCL	10	✓	✓	G	P13	n/a	✓
BLI0083	P	LCL	5	✓	✓	KS *	P16	n/a	✓
BC6325	P	LCL	3	✓	✓	KS	P20	n/a	✓
SC3709	SeV	LCL	4	✓	✓	G	P22	n/a	✓
LNH0108	P	LCL	12	✓	✓	G	P28	n/a	✓
LCA0042	P	LCL	20	✓	✓	G	P30	n/a	✓
LC0209	P	LCL	7	✓	✓	G	P32	n/a	✓
BC6055	P	LCL	1	✓	✓	KS	EBV present	n/a	✓
SC3602	P	LCL	20	✓	✓	KS	EBV present	n/a	✓
LP0584	P	LCL	5	✓	✓	G	P17	✓	✓
LP0582	P	LCL	5	✓	✓	G **	P23	✓	✓
SMA0020	SeV	LCL	5	✓	✓	G	P26	✓	✓
LP0663	P	LCL	3	✓	✓	G	P15	✓	✓
LP0039	P	LCL	4	✓	✓	G	P26	✓	✓
BP6184	P	LCL	14	✓	✓	G	P20	✓	✓
SP3185	P	LCL	12	✓	✓	G	EBV present	✓	✓
LP0225	P	LCL	10	✓	✓	KS	EBV present	✓	✓
SP3277	P	LCL	9	✓	✓	KS	P10	✓	✓
SMA0078	P	LCL	6	✓	✓	KS	P29	✓	✓
SP3068	P	LCL	21	✓	✓	G	P19	✓	✓
SP3154	P	LCL	5	✓	✓	KS	P12	✓	✓
BOX0029	P	LCL	13	✓	✓	G	P24	✓	✓
LPO0036	P	LCL	10	✓	✓	KS	P30	✓	✓
BP6021	P	LCL	6	✓	✓	G	P30	✓	✓
BP6204	P	LCL	5	✓	✓	KS	P19	✓	✓
LP0393	P	LCL	4	✓	✓	KS ***	P25	✓	✓
LP0048	P	LCL	3	✓	✓	G	P20	✓	✓
LP0051	P	LCL	3	✓	✓	G	P19	✓	✓
LP0168	P	LCL	3	✓	✓	G	P14	✓	✓
LCA0042	P	PBL	11	✓	✓	KS ****	n/a	n/a	✓
LNH0108	P	PBL	10	✓	✓	G	n/a	n/a	✓
SMA0078	P	PBL	10	✓	✓	KS *****	n/a	✓	✓
SP3277	P	PBL	11	✓	✓	G	n/a	✓	✓
SP3154	P	PBL	8	✓	✓	KS	n/a	✓	✓

P: Plasmid based reprogramming; SeV: Sendai Virus reprogramming; G: Genome assessed by G-band karyotyping; KS: Genome assessed by KaryoStat™ analysis; * KaryoStat™ assay detected a 133,177 kbp gain on chromosome 12; ** G-band karyotyping detected a balanced translocation between chromosome 4 and chromosome 22; *** KaryoStat™ assay detected a 2071 kbp gain on chromosome 7; **** KaryoStat™ assay detected a 1.70 kbp gain on chromosome 15; ***** KaryoStat™ assay detected a 16,805 kbp gain on chromosome 4.

## Data Availability

Cell lines can be accessed via The Motor Neurone Disease Association by following the guidance and application form included in the following link: https://www.mndassociation.org/research/for-researchers/resources-for-researchers/uk-mnd-collections (accessed on 10 May 2023).

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
