# Peer review of "Generation of an Open-Access Patient-Derived iPSC Biobank for Amyotrophic Lateral Sclerosis Disease Modelling"

_genes, 2023, doi:10.3390/genes14051108_

Round 1

Reviewer 1 Report

This is a mainly descriptive methodological paper, describing the culture methodology used to derive iPSCs from lymphoblastoid cell lines (LCLs) and peripheral blood lymphocytes (PBLs) from healthy volunteers and ALS patients. Then, they describe how to differentiate iPSCs into motoneurons and analyze motoneurons phenotypes. Material and methods are adequate to describe the cell culture steps with accuracy and detail. Then, after describing in the results the derived lines, they demonstrated that one line derived from a patient with FUS mutation reproduced some estructural phenotype: increased accumulation o FUS protein in the cytoplasm and reduction in neurite elongation/complexity. Therefore I have not concerns for the present version, except that suggest that with more about ALS phenotypical expression in iPSCs derived motoneurons, including pTDP43 deposits, the scientific strength of this work would be higher and interesting to study sporadic ALS cellular models.

Author Response

We thank you reviewer 1 for the positive comments. We acknowledge that pTDP-43
staining could strengthen this study. Deep phenotyping of these lines will be achieved
in future projects.

Reviewer 2 Report

Review of manuscript # genes-2376549, “Generation of an open-access patient-derived iPSC biobank for amyotrophic lateral sclerosis disease modelling”

 In the present work, Hedges and collaborators report the creation of an open-access iPSC biobank generated from patients affected by amyotrophic lateral sclerosis pathology and carrying mutations in the TARDBP, FUS, ANXA11, ARPP21, and C9ORF72 genes, alongside healthy controls. As proof of concept, the authors differentiated into motor neurons two iPSC lines carrying FUS p.R521H and p.R522G mutations respectively. These motor neurons accumulated FUS proteins in their cytoplasm as observed in ALS post-mortem tissue demonstrating its interest in disease modeling.

Overall, this open-access biobank is a very important contribution to the ALS research field that will provide useful tools to the labs. The methods and results appear to be accurate and reliable. Using EBV immortalized LCL as a starting point for reprogramming, which could be a problem, is well discussed. Indeed, the opportunity to exploit a pre-existing cell bank with a wide variety of mutations implicated in ALS is understandable. On the other hand, the establishment of iPS lines requires additional quality controls to exclude EBV-positive clones represents a substantial work that is also well discussed.

I have some minor concerns

1/ For figures 3B and 4C, image quantification and image data analysis need to be described in much greater detail, at least the number of neurons or images analysed per experiment.

2/ The introduction could describe shortly the positive effects of Masitinib drug on ALS after Edaravone (line 72) and present the Tofersen which is now FDA-approved to treat SOD1-ALS (line 82)

Author Response

We thank reviewer 2 for the opportunity to strengthen our study.
1) For figures 3B and 4C, image quantification and image data analysis need to
be described in much greater detail, at least the number of neurons or images
analysed per experiment.
We now included the following sentence in material and methods:
Approximately 8-10 fields from two separate wells of a 96-well plate were quantified in
three biological replicates.
And this sentence in Figure 3 caption:
The total number of analysed neurons was; control 1 (8,736), control 2 (7,299), FUS
R521H (6,574), and FUS R522G (6,486).
The following sentence is included in Figure 4 caption:
The total number of analysed neurons was; control 1 (130), control 2 (158), FUS
R521H (238), and FUS R522G (82).
2a) The introduction could describe shortly the positive effects of Masitinib drug on
ALS after Edaravone (line 72)
It now reads:
Co-treatment with Riluzole and the tyrosine kinase inhibitor Masitinib slowed the rate
of functional decline compared to Riluzole plus placebo in a phase 2/3 clinical trial,
indicating that combination therapy may be beneficial in ALS.
2b) and present the Tofersen which is now FDA-approved to treat SOD1-ALS (line
82)
The first ASO for treatment of SOD1-ALS, Tofersen, was recently approved by the
FDA; a milestone in drug development for ALS.
We have also included the relevant references and updated the reference numbering

Reviewer 3 Report

The manuscript of  Hedges et al. summarizes the setup of the ALS iPSC biobank generated from patients carrying mutations in the TARDBP, FUS, ANXA11, ARPP21, and C9ORF72 genes, ALS patients with unknown mutations and healthy controls. Altogether 35 iPSC lines have been generated and characterized. Additionally, iPSCs carrying FUS R521H and R522G and two control lines were differentiated into motor neurons and characterized in detail.

The Introduction shows the benefits of iPSC technology and gives perspective on future studies involving iPSC technology. The methods are described in detail. The Figures are of good quality and illustrate the experiments and workflow well. The potential drawbacks of the study are described.

This is an important manuscript showing an easily accessible resource for the whole scientific community.

Comments:

1. The link to the biobank (Page not found | MND Association)  https://www.mndassociation.org/research/for-research-%20400%20ers/resources-for-researchers/ukmndcollections/   seems not to work properly 

2. It would be useful to summarize shortly acession process and conditions of access

Author Response

Reviewer 3
We thank reviewer 3 the opportunity to clarify these topics.
1) The link to the biobank (Page not found | MND
Association) https://www.mndassociation.org/research/for-research-
%20400%20ers/resources-for-researchers/ukmndcollections/ seems not to
work properly
We have now replaced to a new link in line 405.
https://www.mndassociation.org/research/for-researchers/resources-forresearchers/
uk-mnd-collections
2) It would be useful to summarize shortly acession process and conditions of
access
It now reads (lines 406-407):
Cell lines can be accessed via The Motor Neurone Disease Association by following
the guidance and application form included in the following link:
https://www.mndassociation.org/research/for-researchers/resources-forresearchers/
uk-mnd-collections